# Model-Based Visual Planning with Self-Supervised Functional Distances

**Stephen Tian**[1], **Suraj Nair**[2], **Frederik Ebert**[1], **Sudeep Dasari**[3], **Benjamin Eysenbach**[3],
**Chelsea Finn**[2], **Sergey Levine**[1]
[1]University of California, Berkeley
[2]Stanford University
[3]Carnegie Mellon University

## Abstract

A generalist robot must be able to complete a variety of tasks in its environment. One appealing way to specify each task is in terms of a goal observation. However, learning goal-reaching policies with reinforcement learning remains a challenging problem, particularly when hand-engineered reward functions are not available. Learned dynamics models are a promising approach for learning about the environment without rewards or task-directed data, but planning to reach goals with such a model requires a notion of functional similarity between observations and goal states. We present a self-supervised method for model-based visual goal reaching, which uses both a visual dynamics model as well as a dynamical distance function learned using model-free reinforcement learning. Our approach learns entirely using offline, unlabeled data, making it practical to scale to large and diverse datasets. In our experiments, we find that our method can successfully learn models that perform a variety of tasks at test-time, moving objects amid distractors with a simulated robotic arm and even learning to open and close a drawer using a real-world robot. In comparisons, we find that this approach substantially outperforms both model-free and model-based prior methods. Videos and visualizations are available here: https://sites.google.com/berkeley.edu/mbold.

## 1 Introduction

Designing general-purpose robots that can perform a wide range of tasks remains an open problem in AI and robotics. Reinforcement learning (RL) represents a particularly promising tool for learning robotic behaviors when skills can be learned one at a time from user-defined reward functions. However, general-purpose robots will likely require large and diverse repertoires of skills, and learning individual tasks one at a time from manually-specified rewards is onerous and time-consuming. How can we design learning systems that can autonomously acquire general-purpose knowledge that allows them to solve many different downstream tasks?

To address this problem, we must resolve three questions. **(1)** How can the robot be commanded to perform specific downstream tasks? A simple and versatile choice is to define tasks in terms of desired outcomes, such as an example observation of the completed task. **(2)** What types of data should this robot learn from? In settings where modern machine learning attains the best generalization results (Deng et al., 2009; Rajpurkar et al., 2016; Devlin et al., 2018), a common theme is that excellent generalization is achieved by learning from *large* and *diverse* task-agnostic datasets. In the context of RL, this means we need *offline* methods that can use all sources of prior data, even in the *absence of reward labels*. As collecting new experience on a physical robot is often expensive, offline data is often more practical to use in real-world settings (Levine et al., 2020). **(3)** What should the robot learn from this data to enable goal-reaching? Similar to prior work (Botvinick & Weinstein, 2014; Watter et al., 2015; Finn & Levine, 2017; Ebert et al., 2018b), we note that policies and value functions are specific to a particular task, while a predictive model captures the *physics* of the environment independently of the task, and thus can be used for solving almost any task. This makes model learning particularly effective for learning from large and diverse datasets, which do not necessarily contain successful behaviors.

While model-based approaches have demonstrated promising results, including for vision-based tasks in real-world robotic systems (Ebert et al., 2018a; Finn & Levine, 2017), such methods face two major challenges. First, predictive models on raw images are only effective over short horizons, as uncertainty accumulates far into the future (Denton & Fergus, 2018; Finn et al., 2016; Hafner et al., 2019b; Babaeizadeh et al., 2017). Second, using such models for planning toward goals requires a notion of *similarity* between images. While prior methods have utilized latent variable models (Watter et al., 2015; Nair et al., 2018), $\ell_2$ pixel-space distance (Nair & Finn, 2020), and other heuristic measures of similarity (Ebert et al., 2018b), these metrics only capture *visual* similarity. To enable reliable control with predictive models, we instead need distances that are aware of dynamics.

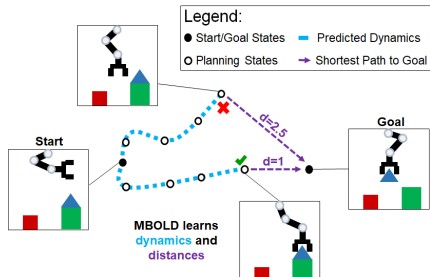

Figure 1: The robot must find actions that quickly achieve the desired goal. State transitions and the true optimal distances between states are unknown, so our method learns an approximate shortest distance function and dynamics model directly on images. These models allow the robot to find the shortest path to the goal at test-time.

In this paper, we propose Model-Based RL with Offline Learned Distances (MBOLD), which aims to address *both* of these challenges by learning predictive models together with image-based distance functions that reflect functionality, from offline, unlabeled data. The learned distance function estimates of the number of steps that the optimal policy would take to transition from one state to another, incorporating not just visual appearance, but also an understanding of dynamics. However, to learn dynamical distances from *task-agnostic* data, supervised regression will lead to overestimation, since the paths in the data are not all optimal for any task. Instead, we utilize approximate dynamic programming for distance estimation. While prior work has studied such methods to learn goal-conditioned policies in online model-free RL settings (Eysenbach et al., 2019; Florensa et al., 2019), we extend it to the offline setting and show that approximate dynamic programming techniques derived from Q-learning style Bellman updates can learn effective shortest path dynamical distances. Although this procedure resembles model-free reinforcement learning, we find empirically that it does not by itself produce useful policies. Instead, our method (Fig. 1) combines the strengths of dynamics models and distance functions, using the predictive model to plan over short horizons, and using the learned distances to provide a global cost that captures progress toward distant goals.

The primary contribution of this work is an offline, self-supervised approach for solving arbitrary goal-reaching tasks by combining planning with predictive models and learned dynamical distances. To our knowledge, our method is the first to directly combine predictive models on images with dynamical distance estimators on images, entirely from random, offline data without reward labels. Through our experimental evaluation on challenging robotic object manipulation tasks, including simulated object relocation and real-world drawer manipulation, we find that our method can outperform previously introduced reward specification methods for visual model-based control with a relative performance improvement of at least 50% across all tasks, and compares favorably to prior work in model-based and model-free RL. We also find that combining Q-functions with planning improves dramatically over policies directly learned with model-free RL.

## 2 RELATED WORK

*Offline and Model-based RL:* A number of prior works have studied the problem of learning behaviors from existing offline datasets. While recent progress has been made in applying model-free RL techniques to this problem of offline or batch RL (Fujimoto et al., 2019; Wu et al., 2019; Kumar et al., 2019; 2020; Nair et al., 2020b), one approach that has shown promise is offline model-based RL (Lowrey et al., 2019; Kidambi et al., 2020; Yu et al., 2020; Argenson & Dulac-Arnold, 2020), where the agent learns a predictive model of the world from data. Such model-based methods have seen success both in the offline and online RL settings, and have a rich history of being effective for planning (Deisenroth & Rasmussen, 2011; Watter et al., 2015; McAllister & Rasmussen, 2016; Chua et al., 2018; Amos et al., 2018; Hafner et al., 2019b; Nagabandi et al., 2018; Kahn et al., 2020; Dong et al., 2020) or policy optimization (Sutton, 1991; Weber et al., 2017; Ha & Schmidhuber, 2018; Janner et al., 2019; Wang & Ba, 2019; Hafner et al., 2019a). **However, the vast majority of**

**these prior works consider the single task setting where the agent aims to maximize a single task reward.** In contrast, in this work we circumvent the need for task rewards by adopting a self-supervised multi-task approach, where a single learned model is used to perform a variety of tasks, specified in a flexible and general way by desired outcomes – i.e., goal images.

*Self-supervised goal reaching*: While the standard RL problem involves optimizing for a task-specific reward, an alternative and potentially more general formulation involves learning a generic goal reaching policy, without task-specific reward labels. In fact, a number of prior works learn goal-conditioned policies using model-free RL (Kaelbling, 1993; Nair et al., 2018; Mandlekar et al., 2019; Nair et al., 2020a), or variants of goal-conditioned behavioral cloning (GCBC) (Ghosh et al., 2019; Ding et al., 2019; Lynch et al., 2020). In our experiments, we show that our method outperforms both model-free approaches and goal-conditioned behavioral cloning. A number of methods combine model-free and model-based elements by planning over a graph representation (Eysenbach et al., 2019; Nasiriany et al., 2019; Savinov et al., 2018; Liu et al., 2020). Such methods can struggle in higher dimensions, where constructing graphs that adequately cover the space may require an excessive number of samples. We compare to these methods in our experiments. Similarly to Finn & Levine (2017); Ebert et al. (2018b); Nair & Finn (2020); Yen-Chen et al. (2019); Suh & Tedrake (2020), our method uses an action-conditioned video prediction model to generate plans. However, these prior methods generally utilize hand-crafted image similarity reward measures such as $\ell_2$ pixel-error (Ebert et al., 2018a; Nair & Finn, 2020) and pixel-flow prediction (Finn & Levine, 2017). In complex scenes, this can become a major bottleneck: predictions degrade rapidly further in the future, making an informative image similarity metric critical for effective planning. We propose to learn *functional* similarity metrics in terms of dynamical distances, which we find can be combined with predictive models to attain significantly improved results.

*Dynamical distance learning*: Our method learns dynamical distances – distances that represent shortest paths – from offline data. In the literature, dynamical distances have been learned via direct regression using online data (Hartikainen et al., 2019), representation learning (Warde-Farley et al., 2018; Yu et al., 2019b), or via Q-learning by relabeling goals (Eysenbach et al., 2019; Florensa et al., 2019). While these last two works are most similar to ours, in that they also employ approximate dynamic programming to learn distances, our method directly combines these dynamical distances with visual predictive models and planning. Lastly, while prior work has also explored combining model-based planning with value functions (Zhong et al., 2013; Lowrey et al., 2019; Hafner et al., 2019a; Schrittwieser et al., 2019; Argenson & Dulac-Arnold, 2020), these works consider the single task domain with a reward function, while our learned value function considers the multi-task goal reaching domain from entirely random, offline data without reward labels.

## 3 THE SELF-SUPERVISED OFFLINE RL PROBLEM STATEMENT

In this section, we introduce notation and define the problem setting. We will employ a Markov decision process (MDP) with state observations $s_t \in \mathcal{S}$ and actions $a_t \in \mathcal{A}$, both indexed by time $t \in 0, 1, \cdots, H$, where $H$ denotes the maximum episode length. The initial state is sampled from an initial state distribution $s_0 \sim p_0(s_0)$, and subsequent states are sampled according to Markovian dynamics: $s_{t+1} \sim p(s_{t+1} \mid s_t, a_t)$. Actions are sampled $a_t \sim \pi(a_t \mid s_t, s_g)$ from a policy that is conditioned on both the current state and a goal state $s_g \in \mathcal{S}$. In our experiments, both the state and goal are images (i.e., $\mathcal{S} = \mathbb{R}^{H \times W \times 3}$).

We tackle offline learning in this setting, assuming access to a fixed dataset $\mathcal{D}$ consisting of trajectories $\{s_0, a_0, s_1, ...s_T\}$ of the agent interacting with the environment. This data can include *any* environment interactions, from expert demonstrations to trajectories which are not particularly successful at any task. In our experiments, we use data collected using a random policy, which is inexpensive to obtain. The agent does not have access to the environment to collect additional training data. Given this dataset, the objective is to determine the optimal goal-conditioned policy $\pi^\star(a_t \mid s_t, s_g)$, under which the agent is able to transition to any goal state $s_g$ from any starting state $s_t$ in the minimum number of time steps possible. Note that unlike in the standard formulation of the RL problem, the agent does not receive any reward signal from its environment.

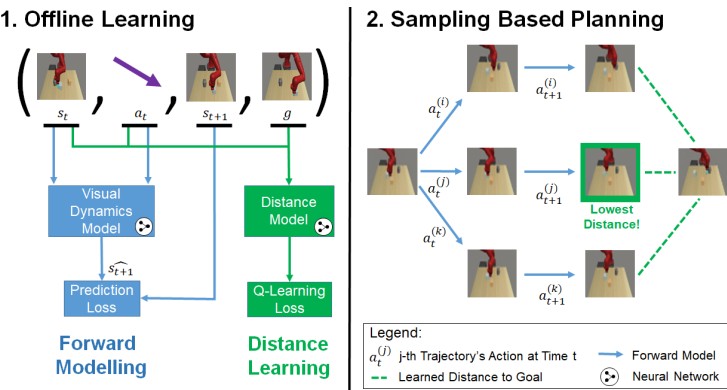

Figure 2: **Model-based visual goal reaching**: *(Left)* During offline learning, we train an image-based predictive model and distance function on the same *random* dataset. *(Right)* At test time, we use the learned distance model for MPC, plugging in the learned distance as a cost function.

## 4 MODEL-BASED VISUAL GOAL-REACHING

In this section we will introduce our method, MBOLD, for offline, goal-conditioned reinforcement learning. MBOLD, illustrated in Fig. 2, is composed of two neural networks: a predictive model and a learned distance function. The video-predictive dynamics model allows the agent to predict the result of hypothetical sequences of actions. However, this model cannot accurately predict far into the future, and has no notion of whether the predicted outcomes are desirable. Thus, we also learn a distance function, corresponding to a value function with a self-supervised goal-reaching reward, which will estimate the timestep length of the shortest path between a predicted state and a given goal. Both networks are trained on the same offline dataset.

At test-time, we use the learned dynamics model and distance function for model-predictive control (MPC). MBOLD predicts future states for candidate action sequences using the learned dynamics model, and uses the learned distance function to determine which action sequence will lead the agent closest to the goal. The first of the actions is then executed, and planning repeats upon receiving the subsequent observation from the environment. The remainder of this section describes how we learn the dynamics model and distance function, and use them to perform control.

**Dynamics learning.** Our method learns environment dynamics in order to solve for actions during test time, *without* an explicit task reward signal during training. MBOLD can use arbitrary image-based forward models, including latent variable models (Hafner et al., 2019b; Lee et al., 2019). The particular choice of model is a design decision when implementing our method. In our implementation, we use a convolutional video prediction model adapted from SAVP (Lee et al., 2018). The network takes as input the current observation $s_t$ and a sequence of $h$ actions $a_{t:t+h-1}$ and returns a prediction for the next $h$ image observations, $\hat{f}_\theta(s_t, a_{t:t+h-1}) = \{\hat{s}_{t+1}, \ldots, \hat{s}_{t+h}\}$. We train this model to minimize the $\ell_2$ image reconstruction loss:

$$\min_\theta \mathbb{E}_\mathcal{D} \left[ \frac{1}{h} \sum_{t'=t}^{t+h} \|\hat{f}_\theta(s_t, a_{t:t+h-1})[t' - t] - s_{t'}\|^2 \right]. \tag{1}$$

**Distance learning.** Our method also learns a dynamical distance function, so that it can evaluate a functional notion of distance from the predicted states to the goal state, for use as a planning cost. However, the environment does not provide a reward signal that might be used to deduce these distances. Indeed, the offline dataset is typically composed of highly suboptimal trajectories, so our method may not even have access to examples of shortest path trajectories between states. Our key observation is that a goal-conditioned Q-function trained on a modified MDP with an indicator cost function yields values that correspond to shortest path distances in the original environment. Thus, Q-learning-like methods can recover optimal distance functions even from sub-optimal data.

We therefore formulate an MDP by augmenting environment trajectories with the reward function $r(s_t, a, s_{t+1}, g) = \mathbf{1}_{s_{t+1}=s_g}$, adding a discount factor of $\gamma$, and considering episodes terminated once they reach the goal state. Note that $s_t$, $s_{t+1}$, and $g$ all represent images, and the reward is

only given when the next state and goal images exactly match. During training, goals are sampled according to a distribution on $\mathcal{S}$, which we will discuss later. If $\gamma < 1$, the Q-values for a policy that maximizes expected *discounted* returns in this MDP can be directly mapped to shortest path distances. Specifically, in discrete state environments, the optimal Q-function can be written as $Q(s, a, g) = \gamma^{d(s,a,g)}$, where $d(s, a, g)$ is a shortest path distance between $s$ and $g$ *after* taking action $a$. Similarly, we can recover $d(s, a, g) = \log_\gamma Q(s, a, g)$. Ultimately, our Q-learning approach corresponds to the following Bellman error optimization objective:

$$\min_\phi \mathbb{E}_{s_t, a_t, s_{t+1} \sim \mathcal{D}, g \sim \mathcal{S}} \left[ Q_\phi(s_t, a_t, g) - (\mathbf{1}_{s_{t+1}=g} + \gamma \mathbf{1}_{s_{t+1} \neq g} \max_{a_{t+1}} Q_\phi(s_{t+1}, a_{t+1}, g)) \right]^2 . \quad (2)$$

In practice, we use a deep network to represent the Q-function. During training, we sample transitions $(s_t, a_t, s_{t+1}, g)$ to optimize the objective in Equation 2. The first three components $(s_t, a_t, s_{t+1})$ can be sampled randomly from the dataset. However, trajectories in the offline dataset may not be directed towards any particular goals, so a key challenge lies in selecting which goals $g$ to choose. The next section describes our approach to sampling these goals.

**Selecting goals for relabeling transitions.** Naïvely choosing $g$, say by sampling random states uniformly from the dataset, will provide an extremely sparse reward signal, as two random state images will almost never be exactly identical. The sparse reward problem can be mitigated by selectively sampling as goals the states that were actually reached in future time steps along the same trajectory as $s_t$ (Kaelbling, 1993; Andrychowicz et al., 2017). More precisely, to sample goals for a transition at time step $t$, we sample a discrete time offset $\Delta \sim \text{Geom}(p)$, where $p \in [0, 1]$ is a hyper-parameter, and use the state at time $t + \Delta$ as the goal. Note that if $\Delta = 1$, the reward for this transition is 1, avoiding the sparsity issue.

However, relabeling *all* transitions in this way creates a major issue: since the distance function would only be trained on goals that were actually reached, it would systematically underestimate the distance to unreachable goals. Put another way, goals that were *not* reached from $s_t$ would be *out-of-distribution* goals for the resulting Q-function. We found this to result in poor performance. In practice, prior work (Kaelbling, 1993; Andrychowicz et al., 2017) actually relabels with a mixture of reached goals and commanded but not necessarily reached goals.

These prior methods can obtain such "negative" goals based on the goals that were commanded during online data collection. This is impossible in our setting, since our offline data may not even have been collected with a goal-directed policy. We therefore need a procedure to select such "negative" goals that are distant yet relevant. Randomly selecting dataset states will lead to pairs of images that are clearly distant with high probability (e.g., pairs in which all objects and the robot have been moved), but not necessarily relevant. We would like a goal sampling procedure that produces less obvious examples of distant states, which are more informative for training. Hard negative mining is one example of such a procedure, where pairs are selected based on the model's predictions, but is computationally expensive with large datasets.

Instead, we build upon the intuition that distance functions are likely to pay excessive attention to fully actuated factors in the state, such as the position of the robot's arm, because they are strongly predictive of distances. We propose sampling "negative" goal states $g$ which have similar actuated components to reached states. When randomly sampling pairs of states under this constraint, the underactuated dimensions (e.g. the objects), which are generally not known, are likely to have distinct positions. Hence, these data points can serve as informative hard negatives that encourage the model to pay more attention to the difficult, underactuated parts of the state. Unlike hard negative mining, this sampling approach is computationally inexpensive, as it does not rely on the current distance function, and practical, as actuated components of the state can typically be measured through encoders on the actuator. In practice, we sample these "negative" goals from observations across all dataset trajectories via nearest-neighbors search, using arm joint $\ell_2$ distance as the similarity key. Note that this does assume proprioceptive state information from the agent (e.g. robot joint angles), which is almost always available in real-world robotics settings, but does *not* require knowledge about object positions or other ground-truth environment information. While we use actuator information for generating training examples, the distance function and dynamics model use only image observations and actions as inputs. See Appendix A.1 for details.

**Control via MBOLD.** At test-time, the learned distance function and dynamics model are used together to solve control tasks via MPC. In other words, the dynamics model predicts how candidate

Figure 3: **Comparative evaluation results:** *(Left)* Example initial states and task definitions for Sawyer object pushing and Franka door sliding simulated environments, as well as the real-world drawer closing task. Note that "hard" tasks require the arm to take detours from moving to the final arm position in order to relocate the object. Arrows indicate successful trajectories. *(Right)* MBOLD is consistently able to outperform prior methods on these harder manipulation tasks, and by a larger margin on the most difficult tasks ("hard" variants of object pushing and door sliding). Error bars show standard deviations over 5 seeds.

actions will affect the environment, and the distance model rates predicted sequences based on which bring the agent closest to the user-defined goal state. This mechanism works as follows: given the current state $s_t$, goal state $s_g$, candidate actions $a_{t:t+h-1}$, and predicted future states $\hat{f}_\theta(s_t, a_{t:t+h-1})$ from the learned dynamics model, the learned distance function calculates

$$V(a_{t:t+h-1}) = \max_\alpha Q_\phi(\hat{f}_\theta(s_t, a_{t:t+h-1})[t+h], \alpha, s_g). \tag{3}$$

In practice, the maximization over $\alpha$ is performed by an actor network learned simultaneously with the Q-function. $V(a_{t:t+h-1})$ acts as an objective function for MPC. Plainly, the controller's goal is to find candidate actions $a_{t:t+h-1}$ which minimize the dynamical distance to the goal $h$ steps into the future. After this process completes, the best action is executed by the agent. Note that this controller re-plans after every action taken in the environment (i.e every timestep), in order to prevent errors in dynamics prediction from compounding.

**MPC Algorithm.** MBOLD uses the CEM algorithm (De Boer et al., 2005) to optimize the objective in Equation 3. It begins by sampling $N$ random trajectories from a prior multi-variate Gaussian distribution. Then, the top $K$ actions which score highest according to $V(a_{t:t+h-1})$ are selected as candidates. A new Gaussian distribution is fit on these candidates, and the loop starts over again by sampling fresh actions from this distribution. After $I$ iterations, the loop finishes and returns the best action found so far. See Appendix A.2 for full CEM implementation details.

## 5 EXPERIMENTS

Our experiments aim to answer three questions: (1) How does MBOLD compare to prior model-based and model-free methods when learning to reach goals from task-agnostic offline data? (2) Can our method perform visual robotic manipulation in real-world settings? (3) How do different dynamical distance learning methods compare to MBOLD in terms of providing effective distance functions for planning?

We first evaluate our method, prior methods, and baselines on three simulated tasks with visual observations: (1) a simple *reaching* task that requires moving a Sawyer 7-DoF arm to a goal location, which provides a way to validate implementations of all methods, (2) *object pushing*, in which a Sawyer arm must relocate an object to a particular goal location, in environments with 1 or 3 objects, and (3) *door sliding*, which requires repositioning a sliding door with a Franka 7-DoF arm. These tasks are challenging because they require long-horizon planning without access to intermediate rewards.

For each task, we define the action space $\mathcal{A}$ such that actions control the Cartesian position of the robot's end-effector, as well as the robot's gripper. We randomly generate a set of 100 test goals, consisting of a goal image and starting state, for each task, on which all methods are tested. A trial is considered successful if the final distance to the goal of each relevant object, e.g. slide position for the door sliding task, ends below a given threshold. For the object relocation task, we evaluate each method on two scenes, containing one and three objects. All evaluation goals require the robot to move one of the objects, with the others serving as distractors. We also study two levels of difficulty: "regular," where goals are generated from random trajectories in which the object moves a certain

Initial image    Final image

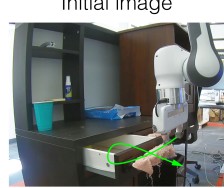 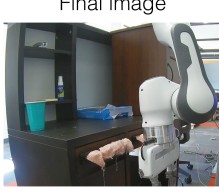

|  | MBOLD (ours) | Visual Foresight ($\ell_2$ pixel error) |
|---|---|---|
| Drawer open | **8/10** | 5/10 |
| Drawer close | **7/10** | 0/10 |

Figure 4: **Real-world robot evaluation:** *(Left)* Third-person view of an example task setting and *(Right)* results. Success rates are computed using 10 trials for each task. Each task is specified by a goal image, and as in previous experiments, the same trained models are used across tasks. Task success is determined by the final position of the *drawer* only.

minimum distance, and "hard," where the arm is additionally enforced to be distant from the object in the goal observation, requiring the robot to push the object and then withdraw the arm. We depict the tasks in Fig. 3 (left) and provide full experimental details in Appendix A.3.

For all tasks, we generate an offline dataset by running random policies for 1e4 episodes of 30 timesteps each. We provide only this offline dataset to all methods, with no online training. At test time, the agent only receives the goal image and current observation at each step, and no intermediate rewards besides those that it computes itself.

**Comparative evaluation.** We compare MBOLD to prior work in model-based and model-free RL. As MBOLD uses purely offline data and does not require rewards from the environment, we make modifications to these methods where necessary to provide a fair comparison. Many of these prior methods (though not all) require the environment to provide a ground truth reward signal. In this case, we provide these methods with simple "uninformative" rewards, following prior work (Nair et al., 2018), which consist of the MSE between the current and goal image. Many of these methods were initially presented in the online setting. The offline setting is harder for RL methods (Fujimoto et al., 2019; Wu et al., 2019; Kumar et al., 2019), partially explaining their poor performance.

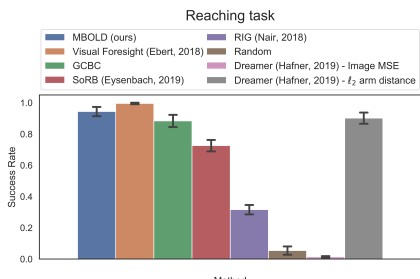

Figure 5: Comparisons on the simple reaching task, where most methods attain good performance.

See Appendix B for details on all baselines. We compare MBOLD to the following methods:

- **Reinforcement Learning with Imagined Goals (RIG) (Nair et al., 2018)**: RIG is a model-free RL method for visual goal-reaching. Unlike the other methods, we still allow RIG to collect additional *online* data to train its policy.

- **Dreamer (Hafner et al., 2019a)**: Dreamer, a model-based method for image-based tasks, also uses a combination of value functions and planning, but uses online data collection and, crucially, ground truth reward signals. We adapt Dreamer for the offline, reward-free setting.

- **Dreamer $\ell_2$ arm distance**: We additionally compare with an "oracle" version of Dreamer that uses privileged information about the ground-truth position of the arm.

- **Search on the Replay Buffer (SoRB) (Eysenbach et al., 2019)**: SoRB performs planning on a graph constructed using learned distances, learned without a reward function.

- **Goal-Conditioned Behavior Cloning**: We train a behavior cloning model using goals sampled from observations achieved further in a given trajectory. This can be viewed as an offline variant of GCSL (Ghosh et al., 2019) or a non-recurrent version of Lynch et al. (2020).

- **Visual Foresight (Ebert et al., 2018b)**: Visual Foresight also plans with an action-conditioned video prediction model, but uses (among other choices) $\ell_2$ pixel error as a cost function.

Since all methods are trained from offline data with no additional environment interaction, we present final performance on the test goals as a bar graph, rather than learning curves. The comparison on the simple reaching task is shown in Figure 5, and suggests that on this task, many of the methods perform quite well. However, on the substantially more complex tasks, shown in Figure 3, we see clearer differentiation between the different algorithms. On harder *object pushing* tasks, MBOLD attains the best performance, by a considerable margin. Interestingly, simple goal-conditioned behavioral cloning actually represents one of the strongest baselines on this task. On the hardest simulated *door sliding* task, our method attains the best performance by a large margin.

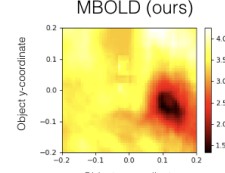
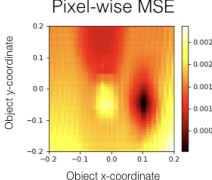
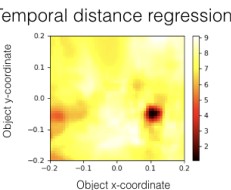

Figure 6: Heatmap visualizations of our distance functions. Each pixel in every heatmap represents the distance between a generated starting image containing the object at that $(x, y)$ coordinate and the fixed goal image (pictured on left). All three distance functions show a minimum when the object position is near the goal position of $(0.1, -0.05)$. However, our Q-function produces a better-shaped signal than the direct regression model, and avoids occlusion errors - like the local minimum at high $y$-values, which plague pixel-wise MSE.

**Real-world evaluation.** We additionally evaluate MBOLD in a real-world drawer manipulation task using a 7-DoF Franka arm. We train the dynamics model and distance function on a preexisting dataset of 1000 trajectories collected by a weakly supervised batch exploration algorithm in prior work (Chen et al., 2020). As shown in Figure 4, MBOLD outperforms visual foresight on both manipulation tasks with visual inputs, particularly on drawer closing, for which simply matching the arm position in the goal image does not solve the task. The success of our method in this domain highlights that our method can be applied to offline datasets collected using different exploration strategies. While MBOLD performs well on manipulation tasks even with complex real-world visuals, we find that the negative sampling procedure we adopt limits precision in matching highly actuated components such as the arm position. We perform additional analysis through simulated experiments detailed in Appendix E.1. Videos of both simulated and real-world task execution can be found at the project website: https://sites.google.com/berkeley.edu/mbold.

**Qualitative analysis.** In this section, we examine the distance functions learned by MBOLD, and show qualitatively that our learned distances better model the dependence of functional separation between two states on the relative positions of objects in their scenes. Figure 6 presents heatmaps of predicted distances for a fixed goal image on the *object pushing* task, as the initial observation is varied based on object position. The robot arm is set to the same position in each initial image. We see that the Q-function is able to learn a relatively well-shaped distance which accounts for the object position.

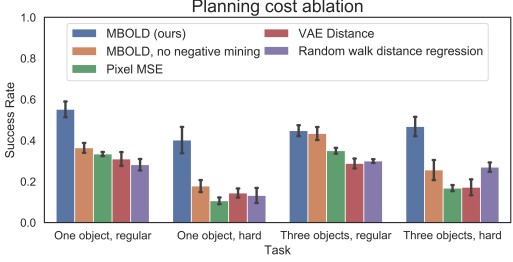

Figure 7: Our learned distance function yields higher success rates than alternative approaches from prior work, such as the $\ell_2$ distance of a VAE latent space (Nair et al., 2018) and temporal distance regression (Hartikainen et al., 2019). We also see consistent improvements from using negative transition mining, especially on "hard" tasks.

We additionally visualize baseline distance models for comparison. First, we look at an ablation of our distance model, which is trained via regression to map pairs of states randomly sampled from a given dataset trajectory to the number of timesteps separating them in that trajectory, and can be viewed as an offline variant of DDL (Hartikainen et al., 2019). We call this scheme that effectively predicts random walk distances "temporal distance regression." The second baseline we compare to is pixel-wise mean-squared error.

We find that the temporal distance regression model produces more sharply peaked distances than the Q-function, and performed worse as a reward signal during planning, as we find through our ablation experiments. The pixel-wise MSE metric produces low distances near the goal object position, but is impacted by occlusions of the objects as well as the position of the visually pronounced arm. While this analysis does not necessarily directly correspond to control performance, as it ignores the movement of the robot, it demonstrates that our learned distances are aware of the functional similarity of nearby object positions, despite the fact that they are learned entirely from images with actions corresponding to the movement of the *arm*, not the *object*.

**Ablations.** Our ablation studies aim to answer three questions: (1) How does Q-learning for learning dynamical distances compare to alternative distance metrics, such as distance in the latent space of a VAE, or dynamical distances learned using direct regression on temporal distances found in random data? (2) How important is mining negative transitions to the performance of our method? (3) How

beneficial is it to combine the learned distance function with planning through a predictive model, as compared to directly acting using the learned policy, as in standard model-free offline RL?

To answer the first two questions, we perform experiments in the *object pushing* domain. We evaluate alternative distance metrics for visual planning, by duplicating the planning setup, using the same dynamics model, and only modifying the metric used for scoring candidate trajectories. The first distance we consider is Euclidean distance in the latent space of a VAE, that is, $d(s, g) = \|e(s) - e(g)\|_2$, where $e$ is a learned encoder, which resembles the reward function used in prior work on image-based goal reaching (Nair et al., 2018). The second is the direct temporal distance regression model described previously. As shown in Figure 7, Q-function distances outperform alternative distances on all of the object pushing tasks. While the temporal distance regression scheme provides competitive performance in some settings, it often provides overestimates of distances between states rather than shortest paths, as shown qualitatively in Figure 6.

We also find that the negative transition mining scheme also consistently improves performance, and is particularly important for the "hard" tasks. We hypothesize this is because augmenting the training data in this way causes learned distance functions to better take into account the positions of objects in the scene, rather than just visually prominent components such as arm position.

To address the third question, we compare our method, which uses learned distances for planning, to the policy discovered when performing Q-learning to learn dynamical distances. As shown in Table 1, the policy learned directly from offline RL alone is greatly outperformed by MBOLD. We hypothesize that this is due to challenges in advantage learning from offline data with extremely sparse rewards.

Table 1: Comparison of success rates $\pm$ standard deviation across 5 random training seeds for our method, which combines Q-functions *and* planning with a model, to a baseline that uses the Q-function to choose actions directly without planning.

|  | Q-function + planning | Q-function only |
|---|---|---|
| 1 object push | $\mathbf{55.2 \pm 4.3}\%$ | $19.2 \pm 3.6\%$ |
| 3 object push | $\mathbf{44.8 \pm 2.9}\%$ | $15.6 \pm 3.6\%$ |
| Reach | $\mathbf{94.4 \pm 3.3}\%$ | $31.8 \pm 5.2\%$ |

## 6 CONCLUSION

We presented a self-supervised approach to tackling goal-reaching tasks, which learns to reach unseen visual goals given only an *offline*, *random* dataset *without reward labels*. Our method combines the strengths of predictive models and learned dynamical distances, where a predictive model can provide effective predictions for planning actions over short horizons, while dynamical distances can provide a useful planning cost that captures distance to goals over longer horizons. By performing visual model predictive control with a learned visual dynamics model and a goal conditioned Q-function as the planning cost, we find that our method is able to perform goal reaching tasks more effectively than model-based planning approaches that utilize other reward specification techniques, as well as purely model-free methods. We show that MBOLD can also scale to real-world manipulation settings and learn from offline datasets collected with various exploration strategies, outperforming visual foresight on a drawer manipulation task. By leveraging offline data collected without a specific goal in mind, our method may make it possible to utilize large, unstructured, open-world robotic manipulation datasets. Scaling up this method to more complex real-world systems and large data sources therefore represents a particularly exciting direction for future work, which may broaden the capabilities and generality of robotic systems.

**Acknowledgements.** We thank students from the Robotic AI and Learning Lab for insightful feedback on earlier drafts of this paper and Aurick Zhou and Danijar Hafner for helpful discussions. This work was supported in part by Schmidt Futures, the Fannie and John Hertz Foundation, the Office of Naval Research (grants N00014-20-1-2675, N00014-16-1-2420, & N00014-19-1-2042), and the National Science Foundation (DGE-1745016 and through an NSF GRFP (GRFP 2018259676)). This research used the Savio computational cluster resource provided by the Berkeley Research Computing program at the University of California, Berkeley.

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

# A MBOLD IMPLEMENTATION DETAILS

## A.1 DISTANCE FUNCTION

This section explains the implementation details for our distance function. Following prior work (Fujimoto et al., 2018), we learn two independent Q-functions and use the *minimum* for performing Bellman backups. Recall that we sampled goals from two distributions: future states in the same trajectories, and states from different trajectories where the robot arm was in a similar position. To implement the second strategy, we fit a $k$-nearest neighbors graph on 200000 (about 60% of total) dataset observations, and use the $\ell_2$ arm joint distance as the similarity key. Each batch contains equal numbers of transitions generated from each goal sampling method. For computational efficiency, we implement the $k$-NN search using the GPU-enabled FAISS library (Johnson et al., 2017).

We relabel half of the transitions in each training batch with reached goals and the other half with "negative" goals with similar actuated components, finding through ablation experiments that this combination achieves stronger performance compared to using just reached goals in our evaluation environments. In other domains, more careful consideration is required to determine if the assumptions which motivate this "negative" goal sampling strategy are satisfied.

We also modify the reward specification scheme by providing a small positive reward at each step where the goal is not reached, and then a large positive reward upon reaching the goal. Specifically, we choose to give a reward of 1 by default and 10 when the goal is reached (compared to 0 and 1 respectively as presented in the discussion in Section 4), although we do not extensively tune this parameter. We find that it does not affect performance in a statistically significant way (results for each reward choice are within 1 standard deviation of one another) to choose this reward over the $(0, 1)$ rewards. Note that this does not change the interpretation of the Q-function as a shortest path distance, merely slightly complicating the conversion calculations from Q-values to distances in timesteps.

Finally, we add an additional loss term to perform conservative Q-learning (CQL) (Kumar et al., 2020), a method for offline model-free RL, which penalizes Q-values of randomly selected actions and increases Q-values of in-dataset actions. We use the Lagrangian version of CQL to automatically tune the weighting term, and detail the parameters below. We find using CQL improves performance on the door sliding task from a mean success rate of 41% to 58%, but does not significantly impact performance on the others.

The Q-function network architecture consists of convolutional and fully connected layers. We define a network called the *convolutional encoder*, which will be used throughout the appendix. This takes as input an image of shape $64 \times 64 \times 6$, containing the starting and goal images concatenated channel-wise, and consists of 4 2D convolutional layers, with $[8, 16, 32, 64]$ filters, respectively, with all with kernel size $(4, 4)$ and strides of $(2, 2)$. We use Leaky ReLU activations after each intermediate convolutional layer, and batch-norm layers after the second and third Leaky ReLUs.

We flatten the output of the convolutional encoder, concatenate the inputted actions, and feed the features through 6 fully-connected linear layers of 128 units each, with the final layer outputting a single value. Each intermediate fully-connected layer is followed by a ReLU activation and a batch-norm layer.

The actor network architecture first contains the above "convolutional encoder", whose outputs are flattened and input into a 10 layer MLP with 128 fully connected units each, and ReLU activations and batch-norm layers in between. The final output, of dimension 4, is passed through a $\tanh$ activation to constrain it to the normalized action space $[-1, 1]$.

Additional training hyperparameters are detailed in Table 2.

## A.2 MODEL-PREDICTIVE CONTROL

In Table 3, we describe the parameters for model-based planning in our experiments. These parameters are shared across all tasks and planning costs (in ablation experiments). Most values are selected based on prior work (Ebert et al., 2018b). We find that replanning every 6 steps produces slightly better performance than replanning every 13 steps, but not by a large margin, and we do

| Parameter | Value |
|---|---|
| Dataset size | 10000 trajectories |
| Train/test/val split | 0.9/0.05/0.05 |
| Trajectory length | 30 steps |
| Observation dimensions | $64 \times 64 \times 3$ |
| State observations in kNN graph | 200000 |
| Goal relabeling sampling parameter ($p$) | 0.3 (tuned over [0.2, 0.3]) |
| Discount factor ($\gamma$) | 0.8 |
| Learning rate | 3e-4 |
| Target network update Polyak factor | 0.995 |
| Batch size | 64 |
| Actor network noise $\sigma$ | 0.1 |
| Actor network maximum noise magnitude | 0.2 |
| Training iterations | 93750 (300 epochs) |
| Optimizer | Adam |
| CQL Lagrange multiplier learning rate | 1e-3 |
| CQL slack parameter $\tau$ (object pushing) | 3.0 |
| CQL slack parameter $\tau$ (reaching) | 3.0 |
| CQL slack parameter $\tau$ (door sliding) | 10.0 |
| CQL number of randomly selected actions | 10 |

Table 2: Hyperparameters for distance learning

not tune this further due to computation constraints. We sample actions using the filtering scheme described in Nagabandi et al. (2020) to make sequences smoother in time. We initialize sampling distributions using each environment's data collection parameters, as shown in Table 4.

To compute the planning cost described in Equation 3, we maximize over $\alpha$ by feeding in the final predicted state to the policy network learned by TD3, and using the outputted action as the maximizer.

| Parameter | Value |
|---|---|
| Planning horizon ($h$) | 13 steps |
| Actions executed per planning step ($k$) | 6 actions |
| CEM Iterations | 3 iterations |
| Elite sample fraction | 0.05 (10 samples) |
| Samples per CEM iteration | 200 samples |

Table 3: Hyperparameters for model-based planning

### A.3 ENVIRONMENTS

The Sawyer environments are adapted from the Meta-World benchmark (Yu et al., 2019a), and the *door sliding* environment is based off of the environment presented by Lynch et al. (2020). For each task, we define the 4-dimensional action space $\mathcal{A}$ such that actions control the Cartesian position of the robot's end-effector, as well as the robot's gripper.

We randomly generate a set of 100 different test goals for each setting. Each task is defined by a goal image and starting state, on which all methods are tested. We define success for each task in terms of the final distance to the goal of each relevant object, e.g. object position for the object repositioning task. A trial is considered successful if the final distance is below a certain threshold $\epsilon$ manually chosen for each task, listed in the table below. We evaluate the success rate of each method over 5 different random training seeds.

We generate offline datasets for each task by running random policies for $1e4$ episodes of 30 timesteps each. In the beginning of each episode, object positions are reset uniformly randomly over the range of possible positions across each joint. The random policy actions are drawn using a filtering technique, which smooths random zero-mean Gaussian samples across time. We apply the

correlated noise scheme described by Nagabandi et al. (2020), setting the hyperparameter $\beta = 0.5$. The parameters of the multi-variate Gaussian samples in each dimension are listed in Table 4.

|  | Reaching | Object pushing | Door sliding |
|---|---|---|---|
| Data colln. stdev ($diag(\Sigma)$) | [0.6, 0.6, 0.3, 0.3] | [0.6, 0.6, 0.3, 0.3] | [0.3, 0.3, 0.3, 0.15] |
| Object compared in success threshold | Arm end effector | Block | Slide |
| Success distance threshold | 0.05m | 0.05m | 0.075m |

Table 4: Environment and task details

## B    COMPARATIVE EVALUATION IMPLEMENTATION DETAILS

### B.1    REINFORCEMENT LEARNING WITH IMAGINED GOALS (NAIR ET AL., 2018)

In this section, we will discuss implementation details of our adaptation of Reinforcement Learning with Imagined Goals (RIG). We begin by training a $\beta$-VAE with latent dimension 8. The VAE is trained on randomly sampled states from the entire offline dataset. For the loss, we use a combination of a maximum likelihood term and a KL divergence term which constrains the latent space to a unit Gaussian. In particular, we compute the mean pixel error, that is, $\frac{1}{HW}\|s - \hat{s}\|_2^2$, where $s$ is the original image, and $\hat{s}$ is the reconstruction, both normalized to be in $[0, 1]$. We add this to the KL divergence between the latent distribution and the unit Gaussian, with a weighting factor of $1e^{-3}$ on the KL penalty.

The architecture of the VAE encoder consists of the "convolutional encoder" described in section A.1, whose features are passed through two FC layers with 128 units with a ReLU activation and batch-norm layer in between. The VAE decoder takes as input latent states into two FC layers with 128 units with a batch-norm layer and ReLU activation after each. This is followed by the inverted architecture of the encoder, consisting of transposed 2D convolutions.

Then, we perform model-free RL in a modified MDP, using encoded observations as a substitute for environment observations, and computing rewards as negative $\ell_2$ distances in latent space. We sample random goals from the multivariate Gaussian prior ($\mathcal{N}(0, I)$) at the beginning of every episode. We use the open-source implementation of soft actor-critic (SAC) in RLKit, and use the default SAC parameters and architecture found in the implementation, making the following modifications: We increase the number of layers of all MLP networks from 2 to 6. We use a maximum path length of 30 steps for consistency with our other experiments, and a discount factor of 0.95. Along with the goal sampled from the prior at the beginning of each episode, we find that relabeling goals with the achieved observation at the end of the trajectory improves performance, and add these transitions to the replay buffer as well. Note that unlike in the original RIG formulation, we do *not* update the weights of the learned VAE using data collected online. We evaluate the learned policy after 600 epochs of training, long after environment returns plateau.

### B.2    DREAMER (HAFNER ET AL., 2019A)

Dreamer, a model-based method for image-based tasks, also uses a combination of value functions and planning. We adapt Dreamer from its original single-task setting to learn a goal-conditioned policy, reward predictor, and value function; however, we do not condition the dynamics model on the goal. Dreamer has been previously demonstrated only in settings where the environment provides rewards to the agent, so we modify the method to learn from unlabeled, offline data by using experience replay. We find that using an indicator reward function as in our method or a heuristically defined reward function, image MSE, causes Dreamer to struggle to learn. We thus additionally demonstrate the performance of Dreamer using a manually specified arm distance reward for the Sawyer reaching task.

We build off of the open source implementation of Dreamer by the original authors, written in TensorFlow2 and found at https://github.com/danijar/dreamer. Specifically, to modify the networks to support goal-conditioning, we add independent convolutional encoders which take the goal image as input to each network. Each encoder consists of 2D convolution layers with [32, 64, 128, 256] filters and kernel sizes of 4 to each network, and we concatenate the flattened features to the inputs

of each network. We additionally increase the number of fully-connected layers for the value and actor networks from 3 and 2 respectively to 10. We use a discount factor of $\gamma = 0.95$. All other hyperparameter values are defaults from the public implementation.

For training, we relabel trajectories sampled from the fixed, offline dataset with a uniformly randomly selected observation from the trajectory as the goal. In most of our experiments, we compute the negative pixel-wise MSE as the reward, but in one reaching experiment, we use the negative $\ell_2$ Euclidean distance between the arm end-effector position and the goal end-effector position. We train for 2000 iterations for each experiment, although initial experiments in which we trained for 20x longer did not yield improved results.

### B.3 GOAL-CONDITIONED BEHAVIOR CLONING

To train a goal-conditioned behavior cloning policy, we begin by relabeling random transitions from the dataset with goals which are later achieved in those trajectories. Specifically, we sample state-goal pairs from trajectories in the dataset by first selecting the initial state index $t_i$ uniformly from all timesteps, and then selecting the goal state index $t_g$ uniformly from timesteps greater than $t_i$. We then train a neural network to predict the transition action $a_i$ given the state $s_i$ and the relabeled goal $s_g$, using a mean-squared error loss.

The network architecture is the same as that of the actor network used in Q-learning for MBOLD, described in Appendix A.1. We train the model for 3125000 iterations (1000 epochs) using a batch size of 32, and use the same optimizer and learning rate as the distance learned for MBOLD.

### B.4 SEARCH ON THE REPLAY BUFFER (EYSENBACH ET AL., 2019)

For Search on the Replay Buffer (SoRB), we train a distributional Q-function to represent distances as in the original paper. Distributional RL discretizes possible value estimates into a set of bins – we use 10 for all of our experiments. We train this distributional Q-function for 300 epochs, as in the distance function training for MBOLD. We also use the same architecture and training scheme, altering the number of outputs to 10 bins and using the KL-divergence loss for the distributional Q-function as in Eysenbach et al. (2019). However, unlike in Eysenbach et al. (2019), we train on just the fixed, offline dataset. We then perform the planning portion of SoRB with the "maxdist" parameter set to 4, after manual tuning. We use a graph size of 2000 states for all experiments, due to computational constraints.

We find that the policy learned through Q-learning performs very poorly at reaching subgoals, so we instead substitute the goal-conditioned behavior cloning policy for this purpose. We find that this greatly improves performance across all tasks.

### B.5 VISUAL FORESIGHT (EBERT ET AL., 2018B)

To compare MBOLD to visual foresight, we use the same dynamics model and planning setup as in MBOLD, however, we substitute the learned dynamical distance function with the $\ell_2$ pixel error cost used in visual foresight.

## C ABLATION EXPERIMENTS IMPLEMENTATION DETAILS

### C.1 VAE DISTANCE

We use the same architecture as the VAE used in the RIG comparison described in Appendix B. We set the latent space dimension to 256 and weight the KL divergence term using a factor of $1e^{-5}$. We train the model for 3125000 iterations (1000 epochs) using a batch size of 32, and use the same optimizer and learning rate as the distance learned for MBOLD.

### C.2 TEMPORAL DISTANCE REGRESSION

To train the temporal distance regression model, we sample state-goal pairs from trajectories in the dataset by first selecting the initial state index $t_i$ uniformly from all timesteps, and then selecting the

goal state index $t_g$ uniformly from timesteps greater than $t_i$. We compute the label for this pair as $\min(t_g - t_i, maxdist)$, where $maxdist$ is a hyperparameter we set to 10. The $maxdist$ parameter helps to improve the optimality of distances on average. We train the neural network to regress this target label using an $\ell_2$ error loss. We train the network for 3125000 iterations (1000 epochs) with a batch size of 32, and use the same optimizer and learning rate as the distance learned for MBOLD.

The architecture for the temporal distance regression model begins with the convolutional encoder described in Appendix B. Its flattened outputs are fed into 5 fully-connected layers of 256 units each, with batch-norm and ReLU activations after each intermediate layer.

### C.3 Q-FUNCTION POLICY

We find that the policy directly learned by our method when learning distances performs extremely poorly. However, performing Q-learning using random shooting over 100 uniformly random actions selected from $[-1, 1]^4$ to optimize over actions to compute target values produces much better results when used directly as a policy, compared to using an actor network to perform this optimization as in our method. Therefore, we report results from acting according to this random shooting method. At test time, we estimate the optimal action $a^\star = \arg\max_a Q(s_t, a, g)$ by again sampling 100 uniformly random actions, and selecting the best one.

## D  COMPUTATIONAL COMPLEXITY ANALYSIS

In this section, we discuss the computation complexity of training and acting using MBOLD.

Training: Training the dynamics model takes about 30 hr while training the distance function takes about 5 hr. These training times are dwarfed by the cost of collecting data in the real world, which could take on the order of 3-4 days in the real world (but can be reused for various tasks). In contrast, a single RL approach only requires learning the distance function. While this means that it takes MBOLD significantly longer to train than the single RL approach, note that the dynamics model can be shared across many tasks. We train the dynamics model for 200k and distance function for 94k training steps. A training step for the dynamics model involves one forward and backward pass through the dynamics model. A training step for the distance function requires sampling positive and negative goals, two Q-function forward passes and a policy network forward pass to compute target values and current Q-values, and a backward pass to update model parameters. In contrast, a single RL approach would just learn the distance function, not the dynamics model. From the above estimates, this means that training steps for the dynamics model are around 3 times slower than training steps for the distance model. Because the dynamics model can be used to perform many tasks, this cost is amortized over these tasks, as compared to a single RL approach.

Acting: Selecting a sequence of actions (6 actions in our experiments) using MBOLD requires one forward pass of the dynamics model for each CEM iteration (3 total in our experiments), and one forward pass through the distance function and policy network. Amortized over a trajectory, this amounts to about 2 seconds wall clock time per action, which can be sped up by around 2x with similar performance by replanning less frequently. For a single RL approach, each action would require just one forward pass through the policy network.

## E  ABLATION EXPERIMENTS

### E.1  NEGATIVE MINING & ACTUATED STATE COMPONENTS

The ablation experiments presented in Section 5 demonstrate that the negative mining technique can improve performance on manipulation tasks, as evaluated by the final position of the object being manipulated. However, in experiments performed in the real-world Franka drawer setting which only required the robot arm to "reach" to a particular location to match the goal, we found that MBOLD achieved a mean final Euclidean distance to goal of 0.14m, while Visual Foresight achieved 0.066m over 10 trials. Here, we conduct additional experiments in simulation to investigate the effect of negative mining on reaching goals based on accuracy of matching the *highly actuated components*, for example, the robot arm. In the single-object block pushing setting, we evaluate the performance of distance functions trained with and without negative mining on reaching the desired

goal arm position. We perform the evaluation using (1) the set of test goals used in our original experiments, which include *object movement*, and (2) an additional set of test goals which only require *robot arm movement*. We present the results in Table 6. We find that training without negative mining improves the planner's ability to reach goal arm positions when goals *also* require object movement, but note that this results in weaker performance in actually relocating those objects, establishing a trade-off. When goals are selected to require just arm movement, performance is comparable with and without the negative sampling scheme.

### E.2 PLANNING HORIZON ABLATIONS

In this section, we investigate the effect of the planning horizon $h$ on control performance. After training distance functions according to Appendix A.1, we perform planning with three different settings for $h$ on the simulated block pushing tasks. We present the results in Figure 8. We find that a longer planning horizon is beneficial, especially for solving more difficult tasks. We hypothesize that this is because longer planning horizons allow the planner and distance function to better distinguish promising predicted states, while the fidelity of state predictions remains relatively high.

### E.3 RANDOM OBJECT RESET ABLATIONS

In this section, we perform experiments to evaluate the impact of the distribution of initial object position on task performance. In particular, we look at the single-object Sawyer pushing task. We collect an additional dataset with the same policy and other parameters as that used in the main comparative evaluations, but restrict the random object initialization position to be within $[-0.05, 0.05]^2$ as opposed to $[-0.2, 0.2]^2$. This represents a 16x reduction in the area of possible initializations. We then train a new dynamics model and distance function from scratch and compare the control performance on the same benchmark tasks from the main comparisons. We present the results in Table 5. We find that the control performance on these tasks remain within one standard deviation despite the restriction in reset position.

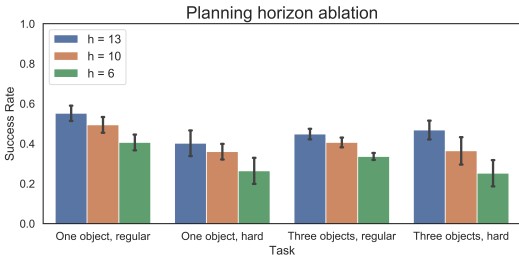

Figure 8: Results for planning horizon ablations.

Table 5: Comparison of success rates for our method when trained using a dataset where object positions at the start of each episode were greatly restricted, compared to uniform over the entire space. Standard deviations are over 5 random seeds.

|  | Uniform reset | Restricted reset |
| --- | --- | --- |
| 1 object push (regular) | $55.2 \pm 4.3\%$ | $54.5 \pm 3.9\%$ |
| 1 object push (hard) | $40.2 \pm 7.2\%$ | $43.2 \pm 7.2\%$ |

Table 6: Effect of training using negative mining on final arm position matching performance. A final $\ell_2$ distance to goal arm position of 0.05m or less is considered a success. Standard deviations of success rates are computed over 5 random seeds.

| Test goals | MBOLD | MBOLD (no negative mining) |
| --- | --- | --- |
| No object movement | $89.2 \pm 1.9\%$ | $91.6 \pm 2.3\%$ |
| Object movement | $64.4 \pm 5.9\%$ | $83.4 \pm 4.0\%$ |

