# OpenReview forum: "Model-Based Visual Planning with Self-Supervised Functional Distances"
_ICLR.cc/2021/Conference — ICLR 2021 Spotlight_

### Official Review · AnonReviewer4 · 2020-10-28
**The paper combines optimal control and reinforcement learning for the execution of robotic manipulation tasks with variable goals. I believe the work may represent a nice contribution to the robotics community.**

**Rating:** 7
**Confidence:** 5

**Review:**

The paper combines optimal control and reinforcement learning (RL) for the execution of robotic manipulation tasks with variable goals. The approach learns, on the one hand, an image-based predictive model using a deep neural network and, on the other hand, a distance cost function using Q-Learning. These model and cost function are used to define the next action to execute using model predictive control (MPC). The cost function is learned from task-agnostic data by randomly generating goal states from the state space and including these goals as input variables in a Q-learning approach. This permits defining cost functions for MPC for variable goals.

The approach is appealing since it exploits the synergies between MPC for planning over short horizons using approximated predictive models and RL for learning longer horizon goals reflected in the cost function.

The approach is an extension of already existing methods that use approximate dynamic programming for distance estimation. One of the claimed contributions with respect to them is the offline nature of the proposed method, in contrast with the online one of these other approaches. This offline nature is highlighted across the paper. In my opinion, going from online to offline is more a downgrade than an upgrade since offline approaches rely on limited and possibly biased data that might be insufficient to guarantee optimality an adaptability to variable scenarios. I would suggest the authors to be more conservative with this particular claim.

The distance learning is based on adding the goal as input variables in a Q-learning setting, where these goals are randomly selected. I am wondering what would be the result if they use a single RL method to learn the policy considering the goal as part of the input space. The authors indeed provide some evidence in the Experiment section (Table I) in this vein by comparing the results of their approach with that of the policy learned in the distance learning method. However, this comparison is still not conclusive. Experiments with more advanced RL approaches (e.g. deep RL) should be provided to fully assess the benefits of combining MPC with RL.

The authors suggest in the Abstract that the approach is suitable for applications where distances in the observation space are not meaningful. However, in the experiments, they use the final distance to the goal of each relevant object to evaluate success and, in distance learning, they use a notion of distance to the goal to define the reward. I would rephrase the statement in the Abstract for consistency.

Provided the method uses limited data in an offline approach, how can the authors guarantee optimality and how can they quantify how different is the learned behavior with respect to the optimal one?

I am also missing an analysis of the computational effort required to learn the predictive model and the distance cost function as well as to define the next action to execute using MPC. I suspect that the computational effort would significantly surpass the one needed by using a single RL approach. This establishes a trade-off between performance and computational effort that needs to be discussed.

The motivation of the proposed approach is very similar to that of inverse reinforcement learning (Fu et al., 2017). The authors should explain the advantages of using their approach with respect to it.

In the second page, the authors refer to their approach as a model-based RL one. I disagree, their approach uses RL only to learn the distance function while the decision-making is carried out using MPC without explicitly representing a policy.

Minor comments:
- The difference between positive and negative goals is not clearly explained (Page 5).
- The bullet points in Page 7 are not properly introduced.


Fu, J., Luo, K., & Levine, S. (2017). Learning robust rewards with adversarial inverse reinforcement learning. arXiv preprint arXiv:1710.11248.

---

> ### Author Response · Authors · 2020-11-16
> **Response to Reviewer 4**
>
> Thank you for the detailed response! We have revised the paper for clarity on the points mentioned and added additional discussion of the limitations of the offline setting and computational complexity. We address each point in greater detail below -- please let us know if these address your concerns.
>
> > going from online to offline is more a downgrade than an upgrade since offline approaches rely on limited and possibly biased data that might be insufficient to guarantee optimality and adaptability to variable scenarios.
>
> In this paper we focused on the purely offline setting for a very practical reason: we had collected a large dataset of real-world experience from a robot and wanted to learn useful behaviors from that data. Prior work such as D4RL (Fu et al, 2020) and RL Unplugged (Gulcehre et al., 2020) has also emphasized the importance of this problem. We agree that online data collection can be preferable when it is feasible; we have added this discussion to the Introduction.
>
> >Experiments with more advanced RL approaches (e.g. deep RL) should be provided to fully assess the benefits of combining MPC with RL.
>
> To clarify, the Q-learning experiments presented in Table 1 are trained using a variant of TD3, a state of the art deep RL method, adapted for offline RL using conservative Q-learning (Kumar et al, 2020). So this is indeed a comparison to a modern (very recent) state-of-the-art deep RL approach.
>
> >The authors suggest in the Abstract that the approach is suitable for applications where distances in the observation space are not meaningful. However, in the experiments, they use the final distance to the goal of each relevant object to evaluate success and, in distance learning, they use a notion of distance to the goal to define the reward.
>
> This was indeed ambiguously phrased -- what we meant to say was that L2 norm (and other hand-crafted metrics) distances in observation (pixel) space are not meaningful. The purpose of learning a dynamical distance function was to acquire a meaningful distance metric in observation space. For evaluation, we used privileged information about the global pose of each object and used the L2 distance between these poses as a metric for success. We emphasize that this object pose information is not provided to our method. We have clarified this phrasing in the Abstract.
>
> >Provided the method uses limited data in an offline approach, how can the authors guarantee optimality and how can they quantify how different is the learned behavior with respect to the optimal one?
>
> This is a good point! Indeed, our method is not guaranteed to recover the optimal policy. The paper currently doesn’t make such a claim, and we are not aware of any optimality guarantees for the deep RL in the offline setting, nor do we attempt to provide any. Since our tasks involve complex dynamics and high-dimensional observations, we cannot analytically compute the reward of the optimal policy for comparison. Our empirical results suggest that MBOLD outperforms a range of prior methods; the fact that MBOLD does not achieve 100% success suggests that there may still be room for improvement.
>
> >I am also missing an analysis of the computational effort required to learn the predictive model and the distance cost function as well as to define the next action to execute using MPC.
>
> We have added a discussion of the computational complexity of both training and acting using MBOLD to Appendix D.
>
> >The motivation of the proposed approach is very similar to that of inverse reinforcement learning (Fu et al., 2017).
>
> Our method and inverse RL solve different problems: our method aims to learn goal-conditioned policies, whereas inverse RL aims to learn policies that imitate expert behaviors. Whereas our approach learns using random data, inverse RL requires data from an expert policy.
>
> > I disagree, their approach uses RL only to learn the distance function while the decision-making is carried out using MPC without explicitly representing a policy.
>
> Per your suggestion, we have revised this reference to say that our method is doing “model-based control.”

---

### Official Review · AnonReviewer1 · 2020-10-28
**Great method, hard to understand**

**Rating:** 7
**Confidence:** 3

**Review:**

# ICLR 2021 Review - Model-based Visual Planning

## Summary

The work introduces a method that allows learning a forward model and a distance function (for visual similarity) from offline data to later use this in planning. The work compares to sensible baselines and achieves competitive results.

## Strengths & Weaknesses

#### Strengths:

- The method is surprisingly simple (in a good way)
- The method can leverage random motion without any task-specific reward assignment to learn a model that's suitable for planning solely based on a goal image.

#### Weaknesses:

- It took me the longest time to figure out what's happening in this method because the paper isn't written terribly clearly. I think this can be resolved by reformulating the "distance learning" part of section 4. The high-level idea is clear from the beginning but I was struggling to see how exactly the distance function is learned. What you're saying is (a) that you assume that for all recorded image observations, you're also storing the robot's proprioceptive joint states, (b) that you're randomly sampling goals in the recorded trajectories, and that (c.) you're learning the distance function based on how similar the joints are, not how similar the images are. Maybe just add to the formula $r(s_t,a_t,s_{t+1},g) = 1_{s_{t+1}=g}$ that this is true if $x_{t+1} \approx g \pm \epsilon$, where $x_{t+1}$ are the joint states of the robot in state $s_{t+1}$ and $\epsilon$ is a small tolerance value. Maybe another diagram to illustrate this would be nice. Figure 2 is nice as a high-level overview but the crucial part is definitely the distance function and I felt that wasn't well-explained.
- I think another important weakness of this method is the scope, i.e. data generation wrt. hard-to-reach areas of the state space. Imagine that your robot arm can't reach an object directly but has to pick up a tool to reach the target object. A random policy may touch the tool but is surely not going to take the tool and figure out that it can be used to reach the target object. This problem can be solved with dense reward (i.e. distance to tool, then distance tool-object) but in this case, this would be incredibly tricky. This is a fundamental problem of using offline data without any annotations and I acknowledge that it wasn't the intention of the method to overcome this exact problem but I'm saying this limits the usefulness of this method in any practical setting where certain areas of the state space are harder to reach.
- No code was included, making this harder to reproduce.

**TL;DR to improve my rating,** it'd be great if you could rewrite the method section to be a bit more clear and maybe add a sentence to the discussion section on how you'd deal with hard-to-reach areas of the state-space. Also please add an anonymized Github repo containing the project code or link a ZIP file - this has become a common practice in many ML conferences.

## Impact & Recommendation

There's a handful of works that deal with offline policy learning but none to my knowledge that learn an image-based distance function that can be used to reach a target configuration. There's VICE-RAQ [1], but they need a lot of positive/negative human labels and there's PlaNet/Dreamer [2] but they do need the reward during training to shape their latent representation. One of my students experimented in the past where they removed the reward prediction loss from PlaNet and the performance of the forward model dropped significantly, indicating that the reward-conditioning is very important there.

I'd recommend accepting this paper and I have high faith that the authors can incorporate my suggestions above.


## Nitpicks and Comments

- Intro paragraph 2 can be shortened significantly. I don't think a lot in there is relevant to set the stage for the paper.
- Make fonts in Fig. 1 bigger. Also in Fig. 1, it looks like the image on the top left is one step before the image at the bottom center. Slightly confusing.
- Fig. 3 fonts on the left are terribly small, also please remove the grey background on the right. Plots don't need to have backgrounds (same for Fig. 4 and 5).
- Please specify the number of seeds for all experiments in the main paper.
- The second paragraph in section 5-"Qualitative Analysis" is also a bit ambiguous and could use some rephrasing.
- Fig. 6 is good.

## References

[1] https://arxiv.org/abs/1904.07854
[2] https://arxiv.org/abs/1912.01603

---

> ### Author Response · Authors · 2020-11-16
> **Response to Reviewer 1**
>
> To address the main concerns in the review, we released our code as a ZIP file in a comment directed to reviewers and area chairs. We also made revisions to the paper text: in Section 4: “Distance Learning”, we improved the clarity of the method description, and in Section 6 we added discussion of the limitations of offline data. Please let us know if these changes address the main concerns in the review. Below, we provided more detailed responses to specific points.
>
> >What you're saying is (a) that you assume that for all recorded image observations, you're also storing the robot's proprioceptive joint states, (b) that you're randomly sampling goals in the recorded trajectories, and that (c.) you're learning the distance function based on how similar the joints are, not how similar the images are.
>
> To clarify the third point, the learned distance function only considers the images, and makes no use of the joints at all. We made edits to clarify this point in Section 4: “Distance Learning” in the paper. The joint states are used only for determining which goals to sample from the recorded trajectories for half of the training transitions, and are not used in any of the reward or loss calculations.
>
> >I think another important weakness of this method is the scope, i.e. data generation wrt. hard-to-reach areas of the state space.
>
> We agree that the availability of sufficiently distributed offline data is a limitation on our method, although, as the reviewer mentions, overcoming this problem is not in the scope of our work. Accordingly, we added a discussion of this limitation in the Introduction.
>
> >rewrite the method section to be a bit more clear and maybe add a sentence to the discussion section on how you'd deal with hard-to-reach areas of the state-space.
>
> Thank you for the suggestion for how to improve the readability of the paper -- we have revised the distance learning section of the paper to more clearly describe how the distance function reward is defined and how goals are sampled for training. We also comment on how we may address the state-space issue in the conclusion section. The edits are made in blue.

---

> > ### Comment · AnonReviewer1 · 2020-11-24
> > **Great improvement**
> >
> > Dear authors,
> >
> > Thanks, I appreciate the changes and comments, and I will update my rating from 6 to 7 since my concerns have been sufficiently addressed.  I find the new version of the paper significantly better readable.
> >
> > I noticed in your code that a lot of files import from the `visual_mpc` package but I couldn't find this anywhere. Is this part of MBOLD or available somewhere?
> >
> > Cheers

---

> > > ### Author Response · Authors · 2020-11-25
> > > **Code clarification**
> > >
> > > Thank you for the feedback! The `visual_mpc` package is an open source package which can be found here: https://github.com/SudeepDasari/visual_foresight. We updated the code ZIP file to include this package -- thanks for pointing this out!

---

### Official Review · AnonReviewer2 · 2020-10-29
**Official Blind Review R2**

**Rating:** 7
**Confidence:** 4

**Review:**

###########################################################################################
Summary:

This paper proposes a novel distance function that considers task dynamics for visual MPC applications.

###########################################################################################

Pros:
1.	The dynamics distance function idea is useful and it can be useful for many existing tasks or visual foresight frameworks.
2.	This work builds on previous works on model-based reinforcement learning and visual foresight. The simple idea of dynamics distance function works pretty well from the results.
3.	This work leverages offline datasets obtained from a random policy instead of online interaction.
4.	The realization of the dynamic distance function idea is interesting by referring to the robot proprioceptive state information and the arm movements. From the experiment, this seems to be a simple idea but works quite well.

###########################################################################################

Cons:
My concern about the generality of this work actually comes from my last point in the Pros section. Will this assumption always true? Or is it just true when the robot takes more pixels in the observation image? Since there is no such analysis or a more detailed explanation about why this helps, it is hard to tell whether this assumption can generalize to other tasks.

###########################################################################################

Recommendation and explanation:

Overall, I feel like this paper presents a simple idea that works quite well for the presented task. I thus recommend acceptance. Please also refer to my Pros section for more details.

---

> ### Author Response · Authors · 2020-11-16
> **Response to Reviewer 2**
>
> Thank you for the review and positive feedback on the paper! With regard to the generality of the work, we found that these assumptions hold in the robotic manipulation domains that we consider. However, we agree with the reviewer that in other domains, this assumption requires more careful thought. We added a discussion of this at the bottom of Section 4.

---

### Official Review · AnonReviewer3 · 2020-11-01
**Review for Model-Based Visual Planning with Self-Supervised Functional Distances**

**Rating:** 7
**Confidence:** 4

**Review:**

#### Summary:

This paper proposes to learn functional distances to varying goals concurrently with a latent dynamics model from images. A network is trained to predict the Q-value of state action pairs for a sparse reward at the goal. This way the Q-function represents the shortest path distance to the goal. The paper also proposes a specific learning scheme for training the model from random rollouts in the environment. The learned distance function is used with the dynamics model for planning from start to goal images using model-predictive control (CEM). The approach is evaluated on simulated reaching and object pushing tasks. It is compared with state-of-the-art methods and demonstrates improved performance in some of the tasks.

#### Strengths:
* The proposed approach for learning functional distance via Q-learning seems novel and effective.
* The approach is evaluated on simulation benchmark tasks and compared with state-of-the-art method. The proposed approach outperforms previous methods in some of the tasks.

#### Weaknesses:
* Training data in the environment is collected using random policies, however, the paper does not provide sufficient detail about the design of these policies. Does data collection require random initial states in the environment and is this feasible in real systems?
* The data collection scheme seems to be important for the performance of the method. Can this be applied to the other state-of-the-art approaches as well, and what is the performance gain for those?
* The paper states that "in theory" MBOLD could use arbitrary image-based forward models. Please detail why and which theory.
* p. 5, the explanation on how state transitions with positive goals from the sample trajectories are generated is not comprehensive ("To sample positive goals..."). Why should it make sense to use goals by images with a \Delta time in between a transition (t -> t+1) and how to obtain this from the data with a given sampling rate? Please clarify.
* The main paper does not give details on how the Q function network is implemented. It only appears in the supplementary material which suggests that the implementation significantly differs from the presented method: The Q function as in eq 3 is actually not implemented by the method, but an actor network directly predicts the best action in a state. Please clarify.
* What is the importance of choice of the prediction horizon h in eq 3 on model performance?
* The training data and goal selection scheme should also be used to train and evaluate the other state-of-the-art approaches to analyze the contributions of training data generation vs. distance learning.
* The training data selection and goal specification encode the tasks in the evaluated simulation environments. How does the method perform with random trajectories and goal states which require less domain knowledge? The claims in the conclusion appear misleading ("By leveraging offline data collected without a specific goal in mind,..."). Please clarify the claim.
* p. 8, Table I, please compare performance with different planning horizons. Does "Q-function only" correspond to a planning horizon of 1 ?

#### Recommendation:
The paper proposes an interesting new approach that learns distances to goal states for model-based visual planning.
The paper should address the points raised in sec "Weaknesses".

#### Question for Rebuttal:
Please address points raised on weaknesses above in the rebuttal.

#### Post-rebuttal comments:
The author response has well addressed most of my concerns on technical details and experiments.

---

> ### Author Response · Authors · 2020-11-16
> **Response to Reviewer 3**
>
> Thank you for your detailed comments and suggestions for how to improve the paper! To address the main points raised in the review, we performed additional experiments analyzing the impact of the planning horizon on control performance, and an experiment exploring the importance of random initial states in the environment. We also revised the paper with several clarifications detailed below.
>
> >p/. 8, Table I, please compare performance with different planning horizons. Does "Q-function only" correspond to a planning horizon of 1 ?
>
> “Q-function only” corresponds to having the agent take actions corresponding to those which maximize a learned Q-function at the current state, without using a forward model.
>
> We performed additional ablation experiments to understand the effect of the planning horizon $h$ on model performance and add them to Figure 7 in Appendix E. We find that a longer horizon length is generally important for better performance, but even with the shortest horizon of 6 timesteps, control performance is competitive with the strongest baseline.
>
> >Does data collection require random initial states in the environment and is this feasible in real systems?
>
> In the experiments presented in the paper, object positions are randomly initialized in the beginning of the episode for all environments. We have revised the description in Appendix A.3 to clarify this.
>
> The use of random initial states is indeed an important consideration for determining our method’s feasibility in real systems. We performed additional experiments to analyze the impact of random initial states on our method, and add them to Table 5 in Appendix F. In the single object pushing domain, we first collected a dataset where the object reset position was restricted to a smaller area of 1/16 the size of the workspace, as opposed to being sampled uniformly over the entire space. We then evaluated control performance on the same benchmark as in our other experiments. We find that for this task, restricting the object resets doesn’t have a statistically significant impact on control performance.
>
> We note that in real world manipulation settings, we have found that roughly random initial states might be obtained using simple scripted routines.
>
> >Can [the data collection scheme] be applied to the other state-of-the-art approaches as well, and what is the performance gain for those?
>
> To ensure a fair comparison, we use the exact same data collection method for all of the methods in our comparison. It’s possible that we do not quite understand your comment: the data collection scheme is largely random, and not very sophisticated (and it is not part of our contribution) -- could you clarify if this random collection scheme is what you are referring to, or something else?
>
> >The paper states that "in theory" MBOLD could use arbitrary image-based forward models. Please detail why and which theory.
>
> This was sloppy phrasing on our part! We have revised the text to indicate more accurately that the particular choice of forward model is a design decision when implementing MBOLD.
>
> >Why should it make sense to use goals by images with a $\Delta$ time in between a transition (t -> t+1) and how to obtain this from the data with a given sampling rate?
>
> As timesteps are discretized throughout our problem formulation, the $\Delta$ time offset used to sample goals is a discrete variable greater than or equal to 1. Therefore, the goal timestep $t_g$ for a transition $t \to t+1$ always satisfies $t_g \geq t+1$.
>
> >The Q function as in eq 3 is actually not implemented by the method, but an actor network directly predicts the best action in a state. Please clarify.
>
> Thank you for pointing out this ambiguity! The maximization over the actions in Equation 3 to compute the value function is performed by an actor network, as we use TD3 (Fujimoto et. al, 2018) to perform Q-learning, and we have updated the paper with this important detail.
>
> >The training data and goal selection scheme should also be used to train and evaluate the other state-of-the-art approaches
>
> In the experiments presented in the paper, the same static dataset of trajectories is used to train each algorithm for each domain.
>
> >How does the method perform with random trajectories and goal states which require less domain knowledge? The claims in the conclusion appear misleading. Please clarify the claim.
>
> To clarify, in our experimental evaluation, we generate training data for each task by having an agent execute a random policy in the environment, and these random policies are not goal-directed. While the goal sampling strategy makes an assumption that the state space contains a highly actuated component as well as less actuated components, which we have updated the paper to discuss at the end of Section 4, we feel that this is a reasonable assumption to make for robotic manipulation environments. We need not make assumptions about the number or positions of objects, for example.

---

### Decision · Program_Chairs · 2021-01-07
**Final Decision**

**Decision:**

Accept (Spotlight)

**Comment:**

The rebuttal (revisions, and released code) very successfully addressed all the major concerns the reviewers had.

Pros: The dynamics distance function is a very neat, simple (which is good in this case) idea that is theoretically sound, has proven to perform well in thorough experimental results, and that can be broadly applied.

Cons: None